# Import Tariff Reduction and Fiscal Sustainability: A Macro-Econometric Modelling for Ethiopia

Gollagari Ramakrishna [1], Berhanu Asefa Gizaw [2], Ch. Paramaiah [3,*], Robinson Joseph [3] and Sania Khan [4]

1   Department of Economics, IBS Hyderabad, IFHE, Hyderabad 501203, India
2   Department of Public Financial Management, ECSU, Addis Ababa P.O. Box. 5648, Ethiopia
3   School of Business, Skyline University College, University City of Sharjah,
    Sharjah P.O. Box 1797, United Arab Emirates
4   Human Resource Management Department, College of Business Administration, Prince Sattam Bin
    Abdulaziz University, Al Kharj 11942, Saudi Arabia
*   Correspondence: channaganu.paramaiah@skylinuniversity.ac.ae

**Abstract:** The main objective of this study is to examine the dynamic impact of import tariff reduction on major macroeconomic variables and its long-run implications for the fiscal sustainability of Ethiopia. In addition, it estimates the increase in the average consumption tax required to compensate for the possible tax revenue loss and fiscal sustainability. A Recursive Dynamic Computable General Equilibrium Model (RDCGEM) is used to assess the dynamic impact on macroeconomic variables of import tariff reduction and examine the increased consumption tax required. The study is novel as there are no studies in general, and for Ethiopia in particular in this regard. We use the available 2013/2014 base year macroeconomic data from national income accounts, fiscal accounts, and balance of payments accounts of Ethiopia. The RDCGEM is calibrated to the base year data, checked for correctness, and tested for robustness using sensitivity analysis. The model then is simulated for the import tariff reduction under the base case and under various tariff reduction scenarios based on which the macroeconomic impacts of tariff reduction are analyzed. Based on the RDCGM outputs, the fiscal sustainability impact of import tariff reduction is verified, and using a cointegration method, we examine the sustainability of fiscal policy. The results suggest a 95 percent import tariff reduction that negatively impacts the major macroeconomic variables. It also leads to long-term fiscal unsustainability. The joint reform of the country has relatively better impacts on major macroeconomic variables but slightly negative effects on household income and consumption.

**Keywords:** tariff reduction; macroeconomic variables; fiscal sustainability; RDCGE; cointegration





## 1. Introduction

The fascination for building macroeconometric models began with the seminal work by Tinbergen [1] and became famous after Klein's publication [2]. Several scholars later used such models for various countries [3]. After that, the macroeconometric methodology underwent several changes, with theory playing an important role and the choice becoming the use of dynamic stochastic general equilibrium (DSGE) models [4]. Macro econometric modeling is extensively used for policy purposes [5]. Boumans et al. [6] provide an excellent historical account of macroeconometric modeling.

Using macroeconometric models in international trade and business has become common these days. The possible effects of imposing a tariff on the macroeconomy of the tariff-imposing country were well discussed in the literature [7]. A tariff is a means of correcting the deficit or disequilibrium in the balance of payments of the tariff-imposing country through the price mechanism. When the scope for the expansion in the country's exports is limited, an import tariff is imposed to limit the imports by making these costlier, which helps bring about a balance in the country's trade account. However, this cannot be

true if the demand for imports in the tariff-imposing country is inelastic, as the imposition of the tariff will not reduce the imports [8].

Ethiopia has been in the process of joining the World Trade Organization (WTO) since 2003, which necessarily required a further liberalization of its international trade regime. This move towards trade liberalization inevitably leads, among other things, to a significant reduction in its tariffs on imports. This study examines the possible impacts of such reduction on the macro economy, particularly on the long-term sustainability of its fiscal policy. We also discuss the level at which the average domestic consumption tax rate needs to be increased to maintain the expected decline in government tax revenue and the sustainability of fiscal policy. Specifically, the study has the following objectives:

1. Examine the impact of import tariff reduction on Ethiopia's imports, exports, current account balance, and government tax revenue.

2. To assess the impact of import tariff reduction on Ethiopia's fiscal sustainability.

3. To measure the extent to which the domestic sales tax is to be raised to compensate for the revenue loss and maintain fiscal sustainability.

The study is novel as no studies are available on the impact of tariff reduction on fiscal sustainability using macroeconometric models, notably in Ethiopia. It helps the policy framers decide on an appropriate tariff policy to achieve fiscal sustainability with possible minimum macroeconomic consequences. The paper is organized into six sections. The literature review is provided in the following section. A discussion of the data and methodology is provided in the third section. Data analysis and findings are presented in the fourth section. A discussion of the results is presented in section five. The final section is on the conclusion and policy implications.

## 2. Review of the Literature

The arguments for and against tariff imposition are well presented in the literature [7]. The imposition of import tariffs to protect domestic infant industries from foreign efficient, well-established industries helps them develop well to face foreign competition. If the tariff is removed, they could compete with foreign enterprises. Accordingly, the tariff can enhance the competitive capability of the protected industries. However, this is only true if the domestic industries efficiently work on advancing themselves rather than taking the tariff as an incentive for their laziness. The protective or production effect of the tariff is the increase in the production of the domestic protected commodity sector, as the imposition of the tariff raises the domestic price of the imported commodity and shifts the demand of the consumers to the domestic commodity/substitutes of the imported commodity. Domestic production of the protected commodity expands as producers are protected from cheap imports. If the demand for the commodity is elastic, the import tariff decreases its consumption. The domestic prices of the commodity rise following tariff imposition, and consumers demand less of the commodity at higher prices. Suppose the imposed tariff is an Ad- valorem tariff. In that case, the imposition of the import tariff increases government tariff revenue from a given unit of the imported item by the amount of the tariff times the value of its imports. However, if the tariff imposition decreases the domestic demand for the given object and its implications, government tariff revenue decreases. The net effect of the impact of imposing a tariff, therefore, is ambiguous.

As the tariff imposition benefits the producers in the protected commodity sector and hurts consumers by decreasing the consumption volume, it transfers consumers' surplus from consumers to producers-which is the distributive effect of tariff and, therefore, plays the role of the income distribution. Taxation is believed to improve the terms of trade of the tariff-imposing country, given that the foreign supply of the protected good could be more elastic. It helps to increase the domestic import price relative to the domestic export supply price. If, however, the supply of foreign supply of the protected commodity is perfectly elastic or if the exporting country is ready to supply the effective demand of the importing country at a constant price, the imposition of tariff will not improve the terms of trade of the tariff-imposing country. The impact of imposing an import tariff and its

removal or reduction by a given country depends on the country's specific situation and trading partner.

A significant amount of literature is available in the context of macroeconometric modeling [9–11]. Akbar and Ahmad [12] present a good review and critique of macroeconometric modeling practices. Acosta et al. [13] discussed two projects undertaken by the Social Science Research Council on constructing a macroeconometric model (1960–1963) and the organization of its conference on quantitative policy analysis in 1963. The projects played an important role in uniting the macroeconometric modeling. They employed econometric models with the collaboration of various officials in the government of the United States. Furthermore, they helped consolidate the work of the committee and government authorities. Another study conducted by the same authors [14] was central in constructing and consolidating the large-scale macroeconometric models as a best and scientific practice in macroeconomics during the 1960s as the bases of the Brookings model (1963-72) and FRB-MIT-Penn model (1966-74). They also discussed the functioning and management of various projects undertaken by the social science research council and its committee on economic stability. They argued that this committee succeeded in producing, collecting, centralizing, and managing data in practice by properly integrating government agencies and academicians.

Pinzón-Fuchs [15] focused on Klein's early trajectory as a student of economics and as an economist (from 1938–1955). They examined the extent to which the people and institutions Klein encountered helped him shape his professional identity. The study identified that macroeconometric modeling became a scientific practice independent of Klein's intentions. Conceptually, macroeconometric modeling is enriched by objectives such as forecasting and policy-level analysis. As Pandit [16] suggested, all the models must satisfy criteria such as theoretical framework, model specification, rich database, and robust methodology.

Aivazian et al. [17] modeled the Russian and Armenian economies by considering various macro indices, such as Gross domestic product, wage payments, exports, and imports of the nations. They employed the cointegration analysis of Engel and Granger [18] and error correction models to identify the choice of predictors of their models. Valdkhani [19] undertook a critical review of macroeconometric modeling in Iran and identified deficiencies and power of these models constructed for Iran as one of the major oil exporting nations earlier. There are two dualities in the macroeconometric models: traditional duality in agro and industrial sectors and oil duality. However, this may not be a deficiency for the other non-oil exporting majors. The author stressed that attention needs to place on capital formation, price, wage, investment, exchange rate fluctuations, unemployment rate, channels of distribution, and demographic characteristics. Bolt et al. [20] studied the global macroeconomic effects of tariffs using a multiregional, general equilibrium model, Euro Area, and Global Economy (EAGLE) model between the United States and China nations. They focused on the macro impact of various trade policy measures and found that the EAGLE model helps analyze the long-run effects of restrictive trade measures on output.

In the African context, Degu and Bekele [21] examined the impact of some macroeconomic factors, including trade openness, inflation, government expenditure, credit extended and foreign direct investment, and natural disaster drought on total factor productivity and its trend in Ethiopia using Time series data spanning from 1991 to 2018. Olofin et al. [22] built a small dynamic econometric model to identify the potential impacts of the monetary policy rate using different scenarios in terms of their possible effects on key macroeconomic indicators such as inflation, exchange rate, output, and lending rate. The macroeconomic realities, such as supply constraints of production activities, the ineffectiveness of interest rates, and fiscal dominance, may lead to monetization of the deficit, inflation, and currency depreciation in several African countries [23].

To assess the macroeconomic effects of tariffs imposed by one country on another, Jacuinot et al. [24] simulated a New Keynesian multi-country model of the world economy by focusing on the United States and Euro Area and the rest of the world. Their results

indicate that tariffs produce recessionary effects in all nations. If the effective lower bound (ELB) ties, tariffs have recessionary effects on the entire Euro area even though they are imposed on these countries. In the same manner, wage effects and elasticity of substitution among tradable variables had a recessionary impact on the Euro area. DSGE models are often used to study economic phenomena involving monetary, fiscal, and trade policy. They are also used to predict and analyse the impacts of shocks such as COVID-19. DSGE models have been the subject of much research in economics, and there have been several developments in this area in recent years. They are helpful in modeling uncertainty, expectations, financial frictions, and heterogeneous agent behavior. These models can be used to study the behavior of the economy, business cycles, and monetary and fiscal policy. However, these models are based on several restrictive assumptions, such as supply-demand equality and agent optimization behavior. A good critique of DSGE models and an alternative model for Africa is provided by Alemayehu and Bewket [25]. Demiessie [26] investigated the impact of the COVID-19 pandemic uncertainty shock on the macroeconomic stability in Ethiopia in the short-run period and the role of fiscal policy using the Dynamic Stochastic General Equilibrium (DSGE) Model.

There are different approaches toward fiscal sustainability [27,28]. If the discounted present value of public debt is stationary and if there is any cointegration relationship between budgetary revenues and expenditures, in the long run, it is widely applied. Afonso [29], using unit root tests for the stock of public debt and cointegration tests between public expenditures and revenues between 1968–1997, found that fiscal policy is only sustainable for some EU countries, except Germany, Austria, and the Netherlands. Similarly, Martin [30] studied US fiscal deficit sustainability based on inter-temporal budget constraints under structural fiscal policy shifts and using a cointegration model with multiple endogenous structural breaks. The results showed that the US budget deficit was sustainable during 1947–1992, with three breaks endogenously determined in the first quarters of 1975, 1985, and 1987. In testing fiscal sustainability using cointegration models, one can incorporate endogenous variables to take care of observed or possible structural breaks.

Wickens and Uctum [31] conducted classical unit roots tests for discounted and undiscounted debt value and found that, generally, the null hypothesis cannot be rejected, only in the case of the Netherlands and Denmark. Unit root test on discounted and undiscounted public debt is one method of testing a country's fiscal sustainability.

Polito and Wickens [32] proposed an index of sustainability calculated as the ratio of the present value of debt, derived from a simple Vector Auto-Regressive (VAR) forecasting model of the economy to the existing level of debt. The authors found that for US and UK sustainability index fluctuated considerably, and in the case of Germany, unification and the European Monetary Union joining sharply deteriorated its fiscal sustainability.

The empirical literature on fiscal sustainability reviewed above employs the use of cointegration analysis and the unit root test methodologies in analyzing the sustainability of fiscal policies using historical data. This study also employs cointegration and unit root test methodologies in assessing the ex-ante impact of import tariff reduction on fiscal sustainability in Ethiopia. However, the study uses the projection data generated by the RDCGE model simulation for the analysis. Moreover, no literature on the ex-ante analysis of import tariff reduction on fiscal sustainability in Ethiopia has been found, such that this study tries to fill the literature gap.

The literature review suggests several studies on macroeconometric modeling but only a few deal with trade-related issues. There are no studies on tariff reduction in general and on Ethiopia in particular. The present study fills this gap as it helps policy framers in Ethiopia assess such a policy's impact while requesting accession to WTO.

### 3. Data and the Model

This study employs macro-level secondary data on different variables from various sources. They include the base year (2013/2014) data on domestic aggregate output, exports,

imports, household consumption, saving, investment, government consumption, and government tax revenue for the different tax accounts (direct tax, sales tax, other indirect tax, and import duty), government debt stock inherited at the base year, on the average interest rate paid on public debt, net foreign saving, net government transfer to households and the rest of the world, primary balance for the base year, net household transfer to the rest of the world, and on other related macro variables. Base year values of other variables are computed using accounting identities through computing averages for the different growth rates used in the study. These data are collected from the Ministry of Finance and Economic Cooperation (MoFEC), the Planning Commission, the Central Statistical Agency (CSA), and the National Bank of the Ethiopian Federal Democratic Republic (EFDR). Other data sources are the World Bank database, the WTO database, and the IMF database.

We use a Recursive Dynamic Computable General Equilibrium Model (RDCGEM) to analyze the dynamic impacts of import tariff reduction on major macroeconomic variables. The model developed for this study is adapted from the static CGE model developed for examining the economy-wide macroeconomic effects of changes in trade policies by Joseph et al. [33] and the dynamic version of Devarajan et al. [34]. We assess the level to which consumption tax should be raised to compensate for the loss of tax revenue from import tariff reduction and maintain the sustainability of fiscal policy in Ethiopia. A cointegration analysis between primary balances and the level of government debt stock is attempted to test the long-term impact on Ethiopia's fiscal sustainability of import tariff reduction in Ethiopia.

Using computable general equilibrium (CGE) models based on ex-ante analysis generally requires using a Social Accounting Matrix (SAM) constructed for the base year period. The level of aggregation or disaggregation of the SAM used could vary depending on the level of analysis intended. It could be a highly disaggregated SAM with multiple sectors and subsectors, factors of production, types of households, etc., or it could be a highly aggregated one with limited sectors and factors of production. Accordingly, an aggregated Ethiopian SAM is constructed for the 2013/2014 fiscal year, a year used as a base year for the analysis based on the availability of the required data.

The RDCGE model parameters and variables are identified and defined. The model parameters and variables are initialized and calibrated to the base year data and are interlinked using economic accounting formulas. The model consists of forty-four variables involving eight parameters, of which two are key parameters, sixteen exogenous variables, and twenty endogenous variables. Model parameters are externally computed or estimated and remain fixed throughout the analysis. Exogenous variables are policy variables and are externally determined, and endogenous variables are internally determined by the model when solved for the values of parameters and exogenous variables at equilibrium.

The correctness of the model is checked using the base year data such that the model should replicate the base year values of the endogenous variables when solved using the base year values of the exogenous variables. This is achieved by comparing the endogenous variables' solution and observed base year values. Finally, the robustness of the model is tested using sensitivity analysis.

Next, the model is simulated under the different import tariff reduction scenarios to analyze the impacts of import tariff reduction on significant macroeconomic variables over the future medium to long-term periods. The model is simulated under three import tariff reduction scenarios: the no policy action or zero percent tariff reduction scenario (base case scenario), 50 percent tariff reduction, and 95 percent tariff reduction scenario. The simulation is conducted, for each tariff reduction scenario, by exogenously changing the values of the exogenous dynamic variables each year (assuming the change is at the start of the year and the impact is measured at the end of the year) and letting the model solutions for the endogenous variables (at the end of the year). The model is also used to assess the level to which the domestic average consumption tax rate (defined in the model as the average of all indirect tax rates) should be raised to compensate for the tax revenue loss and maintain the sustainability of fiscal policy in Ethiopia.

Fiscal sustainability is verified by testing the cointegration [18] between the primary balance share in GDP and one year-lagged value of the public debt stock share in GDP. This is achieved by constructing and estimating a fiscal reaction function. The fiscal reaction function expresses the current primary balance (the difference between revenue and expenditure) as a function of the one-period-lag value of public debt and other possible explanatory variables. Accordingly, the primary balance mainly depends on the initial public debt stock (inherited at the base year) and on the time series values of fiscal revenue and expenditure data obtained from the RDCGE model simulation over the future reasonably long-term period.

The RDCGE model has the equations categorized into five blocks. The first block comprises equations specifying the real flows of commodities. The nominal flow equations are specified under the second block, while the price equations are specified under the third block. In block four, equations defining equilibrium conditions are specified, while equations defining the model's dynamism are specified under block five.

### 3.1. Block 1: The Real flow Equations

Output aggregation equation (export transformation equation)

It is assumed that there are two domestic production sectors. One is a tradeable sector that produces products supplied only to the export market. The second sector is the non-tradable sector that produces the products only for the domestic markets and is consumed domestically. The total domestic output is aggregated using the product or export transformation equation provided below.

$$X = \alpha_t \left( \delta_t E^{\rho_t} + (1 - \delta_t).Ds^{\rho_t} \right)^{1/\rho_t} \tag{1}$$

where

$X$ = The total/aggregate domestic product supplied (at factor cost) at time t. It is the aggregation of the domestic output produced for domestic market supply and the domestic output produced for export. The function is also called a product aggregation function.

$\alpha_t$ = The export transformation efficiency or scale parameter.

$E$ = The quantity of domestic output supplied to world export markets.

$D_s$ = The quantity of the domestic output supplied to domestic markets for domestic consumption.

$\delta_t$ = Share parameter in export transformation equation denoting the share of export volume in the total volume of domestic supply.

$\rho_t$ = Export transformation parameter (the parameter for the transformation function).

The function is assumed to be concave and is specified as a constant elasticity of transformation (CET) function with transformation elasticity ($\sigma_t$) equals to $1/(\rho_t - 1)$.

On the other hand, the value of domestic output equals the sum of the value of domestic output supplied to the export market, and the value of domestic output supplied to the domestic market, as presented in Equation (1.1) below.

$$Px.X = Pe.E + Pd.Ds \tag{1.1}$$

where

$Px$ = the average supply price of the domestic product at time t,

$Pe$ = the export supply price of export goods,

$Pd$ = the domestic market supply price of the producer.

### 3.1.1. Export Supply

Using the first-order conditions, the optimal combination of the export and domestic supply can be obtained from Equations (1) and (1.1). The optimal combination of the quantity of domestic output produced for domestic market supply and the amount of output domestically produced for export markets is expressed as a function of the ratio

of the respective prices and the elasticity of export transformation. The optimal solution, described as the export supply equation, is presented using Equation (2) below.

$$E = \left( \frac{(1 - \delta_t)}{\delta_t} \right) \cdot \left( \frac{Pe}{Pd} \right)^{\frac{1}{\rho^t - 1}} . Ds \tag{2}$$

### 3.1.2. Composite Commodity Supply

In addition to the domestic market supply of the domestically produced output that is domestically demanded, the domestic market is supplied with outputs imported from world import markets. The aggregate output supplied to the domestic market is called a composite commodity. The quantity of composite commodity supplied is obtained using the Armington aggregation function [34] as presented by Equation (3) below.

$$Qs = \alpha_q \left( \delta_q M^{-\rho_q} + (1 - \delta_q) Dd^{-\rho_q} \right)^{-1/\rho_q} \tag{3}$$

where

$Qs$ = Quantity of composite commodity supplied to the domestic market. It is also called the import substitution function/equation.

$M$ = Quantity of imported commodities supplied in the domestic market.

$Dd$ = Quantity of demand for domestically produced commodities in the domestic market.

$A_q$ = The substitution efficiency in the Armington function.

$\rho_q$ = The Armington parameter in the composite product supply function.

Equation (3) defines an Armington function to aggregate a domestic market composite commodity supply. Assuming that imports and domestic goods are imperfect substitutes [22]; the composite commodity is given by a constant elasticity of substitution (CES) aggregation function of import (M) and the domestically produced commodity demanded in the domestic market (Dd), with import substitution elasticity $(\sigma_q)$. The import substitution elasticity can be expressed in terms of the Armington parameter as: $1/\left( \rho_q + 1 \right)$.

The value of the composite commodity equals the value of the imported goods, and the value of domestic products supplied to the domestic market can be provided by Equation (3.1) as below.

$$Pq.Qs = Pm.M + Pd.Dd \tag{3.1}$$

where

$Pq$ = The composite commodity supply price

$Pm$ = The import supply price

$Pd$ = The domestic good supply price

### 3.2. The Import Demand

The optimal combination of the demand for domestic goods and the demand for import goods in the domestic market can be solved from the first-order conditions of Equations (3) and (3.1) provided above. The optimal quantity of import supply is solved and provided by Equation (4) below.

$$M = \left( \left( \frac{Pd}{Pm} \right) \cdot \left( \frac{\delta_q}{(1 - \delta_q)} \right) \right)^{1/(1 + \rho_q)} . Dd \tag{4}$$

### 3.3. The Aggregate Domestic Demand

The aggregate quantity of composite commodity demanded in the domestic market is the sum of the quantity demanded by private consumption, the quantity demanded

by aggregate investment, and the quantity demanded by government consumption. This domestic aggregate demand equation for the composite commodity is given below.

$$Qd = Cn + Z + G \tag{5}$$

where

$Qd$ = total quantity demanded in the domestic market,
$Cn$ = quantity demanded by the household at the time,
$Z$ = aggregate investment demand,
$G$ = government consumption demand at the time.

### 3.4. Household Consumption

Households spend on purchasing consumption goods using their real income, that is net of tax payment and savings. The quantity of private consumption can, therefore, be given as below.

$$Cn = \left(1 - t_y - s_y\right) . \left(\frac{Y}{Pt}\right) \tag{6}$$

where

$Y$ = the nominal income of a household at time t,
$Pt$ = the sales price of composite goods in the domestic markets,
$t_y$ = the average tax rate,
$s_y$ = the average saving rate by households.

### 3.5. Block 2: The Nominal Flow—Household Income

Household nominal income is the sum of income from the sales of productive resources, transfers from the government, and remittances from non-residents. This is given in Equation (7) below.

$$Y = Px.X + Pq.tr + Er.re \tag{7}$$

where

$Y$ = nominal household income,
$tr$ = government transfer to households,
$re$ = remittance received by households from foreign residents.

#### 3.5.1. Aggregate Saving

Aggregate nominal savings are the sum of household, net foreign, and government savings. This is given by the following equation (Equation (8)).

$$S = s_y.Y + S_g + B \tag{8}$$

where

$S$ = is total aggregate saving,
$S_y$ = is the household saving (proportion of income saved by households),
$Sg$ = is government saving,
$B$ = net foreign saving.

#### 3.5.2. Government Saving

The government saving ($S_g$), on the other, is the difference between its total revenue and its current expenditure, which is given by Equation (9) below.

$$S_g = Tax - G.Pt - Pq.tr + Er.ft \tag{9}$$

where

$Tax$ = government tax revenue and
$tr$ = a net transfer from the government to households

ft = a net official transfer (foreign grant)

### 3.5.3. Government Tax Revenue

On the other hand, the total government tax revenue is the sum of import tariff revenue, export tax revenue, sales tax revenue, and income tax revenue. This is given by Equation (10) below.

$$Tax = t_m.wm.Er.M + t_e.Pe.E + t_s.Pq.Qd + t_y Y \tag{10}$$

where

$t_m$ = average import tariff
$wm$ = world price of import goods
$Er$ = exchange rate
$t_e$ = average export tariff rate

### 3.6. Primary Balance

The government budget gap or primary balance is given by the difference between government saving and government capital investment expenditure and is presented in Equation (11) below.

$$Pbt = S_g - I_g \tag{11}$$

### 3.6.1. Net Foreign Saving

Net-foreign saving (Balance of payment) is the difference between income and transfer from rest-of-the world countries and that is paid or transferred to the rest-of-the-world countries. This is given by Equation (12) below.

$$B = Pm.M - Pe.E - re - ft \tag{12}$$

where

$B$ = is net foreign saving

### 3.6.2. Block 3: The Price-Exchange Rate Equation

As the data will be converted and analyzed in terms of foreign currency units, the Exchange Rate will be normalized and set to unity in the model. This is given by Equation (13) below.

$$Er = 1 \tag{13}$$

### 3.6.3. Domestic Import Price Equation

The domestic unit market price of imported goods is a function of the world unit price of imports in the world import market, the average import tariff, transaction costs, and the exchange rate. However, as the model assumes the absence of transaction costs, the domestic price of imported commodities is expressed as a function of only the world import price, the average import tariff, and the exchange rate as given in Equation (14) below.

$$Pm = wm.(1 + t_m).Er \tag{14}$$

where

$Pm$ = Domestic unit price of imported commodity,
$w_m$ = World unit price of the imported commodity in Foreign Currency Unit (FCU),
$t_m$ = Average import tariff rate on imports,
$Er$ = Nominal exchange rate in Local Currency Unit (LCU) per Foreign Currency Unit (FCU)

### 3.6.4. The Export Supply Price

The domestic export supply price is a function of the world of export price, the average export tariff, and the exchange rate. The domestic price of imports is directly proportional to the world export price and the exchange rate, while inversely related to the average export tariff. Accordingly, the domestic export supply price is given using Equation (15) below.

$$Pe = we.Er/(1 + t_e) \tag{15}$$

where

$Pe$ = Domestic export supply price,
$W_e$ = World unit price of the exported commodity in Foreign Currency Unit (FCU),
$t_e$ = Average export tariff rate on imports,
$Er$ = Nominal exchange rate in Local Currency Unit (LCU) per Foreign Currency Unit (FCU)

### 3.6.5. The Unit Price of Domestic Aggregate Output

The unit price of domestically produced output (the average production unit price of domestic output) is a function of the domestic export supply price, the quantity of export, the domestic market supply price of domestically produced output, the quantity of the domestic market supplied domestically produced output and the quantity of the aggregate output. The aggregate output price is directly related to the export supply price, the quantity of exports, the domestic market supply price of the domestically produced output, and the quantity of the domestic market supply of the domestically produced output. The average domestic output price (also called average unit factor cost) is given in Equation (16) below.

$$Px = (Pe.E + Pd.Ds)/X \tag{16}$$

where

$P_x$ = Average aggregate output price (production price)
$Pe$ = Domestic export supply price,
$P_d$ = Domestic market supply price of domestically produced output
$E$ = Quantity of export
$X$ = Quantity of aggregate domestic output
$Ds$ = Quantity of domestic market supply of domestically produced output

### 3.6.6. The Unit Composite Commodity Supply Price Equation

The composite commodity is the aggregate of commodities supplied to domestic markets. It is the aggregate of imported commodities, and domestic market supply of domestically produced output aggregated using the Armington aggregation function. Accordingly, the unit composite commodity supply price is a function of the domestic price of imports, the domestic market price of the domestically produced commodity, the quantity of imports supplied, the quantity of domestically produced commodity supplied to the domestic market, and the quantity of the composite commodity supplied. The composite supply price is given by Equation (17) below.

$$Pq = (Pm.M + Pd.Qd)/Qs \tag{17}$$

where

$Pq$ = Composite commodity supply price,
$P_m$ = The domestic market price of imports
$M$ = Quantity of imports
$P_d$ = The domestic market price of the domestically produced commodity
$Q_d$ = Quantity demanded composite commodity in the domestic market
$Q_s$ = Quantity supplied of the composite commodity

### 3.6.7. The Final Sales Price (Consumer Price)

The final sales price (the consumer price) of the composite commodity is a function of the average sales tax and the composite commodity supply price. It is the sum of the composite commodity supply price and the percentage of the composite commodity supply price paid by consumers as sales tax, depending on the existing sales tax rate. The equation of the final sales price is given by Equation (18) below.

$$Pt = (1 + t_s).Pq \tag{18}$$

### 3.6.8. Block 4: The Equilibrium

The quantity demanded is equal to the quantity supplied both for domestic products and for the composite products in the domestic market. Equation (19) describes that the quantity demanded is equal to the quantity supplied for the domestically produced goods.

$$Dd = Ds \tag{19}$$

Similarly, the quantity demanded is equal to the quantity supplied for composite products in the domestic market. This is given by Equation (20) below.

$$Qd = Qs \tag{20}$$

### 3.6.9. Block 5: The Dynamic Equations

The dynamic equations are not part of the RDCGE model equations that are simultaneously solved but equations that determine the values of the dynamic exogenous variables. The RDCGE model provides optimum solutions for endogenous variables by solving simultaneous equations. There are three dynamic equations provided below.

The government capital investment expenditure ($I_g$)

The government capital investment expenditure as a share of GDP had been growing with an average annual growth rate of 2 percent from 2000/2001 to 2013/2014. It is assumed that the government expenditure as a share of GDP continues to grow with this average rate over the future period included in the analysis.

Accordingly, the value of the government capital investment expenditure in the future t years following the base year is given by Equation (21) below.

$$Ig_t = Ig_0.(1 + g_{ig})^t \tag{21}$$

where $I_t$ is government capital investment share in GDP at time t, $Ig_0$ is government investment GDP share at base year, and $g_{ig}$ is the average annual growth rate of the share of government investment in GDP.

### 3.6.10. The Government Consumption Expenditure (G)

The government consumption expenditure as a share of GDP is one of the exogenous policy variables. The share of government consumption expenditure in GDP had been declining by an annual average of 3 percent from 2000/2001 to 2013/14. It is assumed that the share continues to decline annually for the entire period included in the study. The value of government consumption expenditure as a share of GDP at any time t is given by the following Equation (22).

$$G_t = G_0(1 + g_g)^t \tag{22}$$

where $_{It}$ is government consumption expenditure t years after the base year, $G_0$ is government expenditure at the base year, $g_g$ is the average declining rate of government's consumption expenditure as share of GDP.

### 3.6.11. Domestic Output at Factor Cost (X)

The domestic output as a share of GDP had been increasing by an average of 1 percent over the 2000/2001–2013/2014 period and is assumed to continue growing by this average rate over the analysis period. The volume of the domestic output as a share of GDP at any time t after the base period can, therefore, be given using Equation (23) below.

$$X_t = X_0(1 + g_x)^t \tag{23}$$

where $X_t$ is the quantity of domestic output GDP share at time t after the base period, $X_0$ is the volume of the output-GDP share at the base period, $g_x$ is the average rate of annual growth.

### 4. Data Analysis and Findings

*Macroeconomic Impact Analysis*

It is well discussed in the methodology section that the Recursive Dynamic Computable General Equilibrium model is employed in analyzing the dynamic impacts on macroeconomic variables of import tariff reduction. The analysis is achieved through initialization and calibration of model variables and their base year values, checking the accuracy of the RDCGE model in fitting to the base year macroeconomic data, checking the robustness of the model through sensitivity analysis, conducting simulation under the different import tariff reduction scenarios and interpreting the results.

### 5. Calibration of Model Variables and Parameters

The model variables and parameters are initialized to their base year observed values or calibrated to their derived base year values using the calibration equations. These are presented in the Tables 1–3. The initialization and calibration are performed mainly using the macro-SAM constructed for the base year.

**Table 1.** Initialization and calibration of exogenous variables to base year data.

| Variable | Calibration Formula | Base Year Value |
|---|---|---|
| World import price (wm0) | =Pm0/Er0*(1 + tm0) | 0.953 |
| World export price (we0) | =Pe0*Er0/(1 + te0) | 1.000 |
| Average import tariff rate (tm0) | =SAM(R6,E2)/SAM(R10,E2) | 0.050 |
| Average export tariff rate (te0) | =SAM(R7,E2)/SAM(R1,E11) | 0.000 |
| Average sales tax rate (ts0) | =(SAM(R5,E2) + SAM(R8,E2))/Qs0 | 0.061 |
| Average direct tax rate (ty0) | =SAM(R9,E3)/Y0 | 0.046 |
| Average household saving rate (sy0) | =(Y0-Pt0*Cn0-ty0*Y0)/Y0 | 0.238 |
| Government consumption (G0) | =SAM(R4,E2)/Pt0 | 0.067 |
| Government transfer (tr0) | =SAM(R3,E4)/Pq0 | -0.009 |
| Net foreign transfer (ft0) | =SAM(R4,E11)*Er0 | 0.021 |
| Net private remittance (re0) | =SAM(R3,E11)*Er0 | 0.071 |
| Government capital expenditure (Ig0) | | 0.087 |
| Domestic output (X0) | =SAM(R3,E1) | 0.103 |

Source: Computed.

Accordingly, the average annual growth rate of the domestic output-GDP ratio is computed to be 1 percent, the average annual government capital expenditure-GDP ratio growth rate to be 2 percent, and the average annual government consumption expenditure-GDP ratio growth rate to be -3 percent. These annual growth rates are assumed to remain constant over the period covered in the analysis.

**Table 2.** Initialization and calibration of endogenous variables to the base year data.

| Variable | Calibration Formula | Base Year Value |
|---|---|---|
| Export (E) | = SAM(R1,E11) | 0.118 |
| Import (M0) | =SAM(R11,E2) + SAM(R6,E2) | 0.312 |
| Domestic good supply (Ds0) | =SAM(R3,E1)-SAM(R1,E11) | 0.799 |
| Demand for domestic supply (Dd0) | =Ds0 | 0.799 |
| Composite commodity supply (Qs0) | =M0 + Dd0 | 1.111 |
| Composite commodity demand (Qd0) | =Qs0 | 1.111 |
| Tax revenue (Tax0) | =tm0*wm0*Er0*SAM(R11,E2)+te0*Pe0*E0+ts0*Pq0*Qd0+ty0*Y0 | 0.128 |
| Household Consumption (Cn0) | =SAM(R3,E2)/Pt0 | 0.661 |
| Total household income (Y0) | =Px0*X0+Pq0*tr0+Er0*re | 0.980 |
| Aggregate saving (S0) | =SAM(R10,E3) + SAM(R10,E4)+SAM(R10,E11) | 0.406 |
| Domestic import price (Pm0) | =1 | 1.000 |
| Export supply price (Pe0) | =1 | 1.000 |
| Composite commodity sales price (Pt0) | =Pq0*(1 + ts0) | 1.061 |
| Output price (Px0) | =1 | 1.000 |
| Price of domestic good (Pd0) | =1 | 1.000 |
| Exchange rate (Er0) | =1 | 1.000 |
| Aggregate investment (Z0) | =SAM(R2,E10)/Pt0 | 1.000 |
| Government saving (Sg0) | =Tax0-G0*Pt0-Pq0*tr0+Er0*ft | 0.383 |
| Budget gap (pbt0) | =Sg0-Ig0 | 0.086 |

Source: Computed.

**Table 3.** Initialization and calibration of parameters.

| Parameters | Calibration Formulas | Value |
|---|---|---|
| Elasticity of export transformation ($\sigma_t$) | =previous studies and Guesstimation | 80.0 |
| Elasticity of import substitution ($\sigma_q$) | =previous studies and Guesstimation | 0.5 |
| Rho for export transformation ($\rho_t$) | =$(1/\sigma_t) + 1$ | 1.01 |
| Scale parameter for Export transformation ($\alpha_t$) | =$X0/(\delta_t*E0^{\wedge}(\rho_t)+(1-\delta_t)*Ds0^{\wedge}(\rho_t))^{\wedge}(1/\rho_t)$ | 2.01 |
| Share parameter for export transformation ($\delta_t$) | =$1/(1+(Pd0/Pe0)*(E0/Ds0)^{\wedge}(\rho_t - 1))$ | 0.51 |
| Armington parameter ($\rho_q$) | =$(1/\sigma_q)-1$ | 1.00 |
| Scale parameter for Armington ($\alpha_q$) | =$Qs0/(\delta_q*M0^{\wedge}(-\rho_q) + (1-\delta_q)*Dd0^{\wedge}(-\rho_q))^{\wedge}(-1/\rho_q)$ | 1.68 |
| Share parameter for Armington ($\delta_q$) | =$1/(1+(Pd0/Pe0)*(E0/Ds0)^{\wedge}(\rho_q - 1))$ | 0.13 |

Source: Computed.

## 6. Discussion

We have used a recursive dynamic computable general equilibrium model to assess the impact of tariff reduction on macroeconomic variables and fiscal sustainability. Before we discuss the results, we present the model performance.

### 6.1. Model Accuracy

To check whether the model fits the Ethiopian economy using the SAM, the model is solved under the base case scenario (or the no policy change scenario) using the values of the exogenous variables and the values of the model parameters for the base year. The

accuracy check is performed by comparing each endogenous variable's observed base year values with their respective solution values. The test result is presented in Table 4.

**Table 4.** Sensitivity test.

| Variable Name | Original Value | Final Value 50% Reduction (1st Scenario) | Final Value 50% Increase (2nd Scenario) | Comparison | | |
|---|---|---|---|---|---|---|
| | A | B | C | D = B/A | E = C/A | F = C/B |
| E | 0.118 | 0.119 | 0.118 | 1.00 | 1.00 | 1.00 |
| M | 0.313 | 0.313 | 0.312 | 1.00 | 1.00 | 1.00 |
| Ds | 0.799 | 0.800 | 0.800 | 1.00 | 1.00 | 1.00 |
| Dd | 0.799 | 0.800 | 0.800 | 1.00 | 1.00 | 1.00 |
| Qs | 1.111 | 1.113 | 1.112 | 1.00 | 1.00 | 1.00 |
| Qd | 1.111 | 1.113 | 1.112 | 1.00 | 1.00 | 1.00 |
| TAX | 0.128 | 0.128 | 0.128 | 1.01 | 1.01 | 1.00 |
| Cn | 0.661 | 0.662 | 0.661 | 1.00 | 1.00 | 1.00 |
| Y | 0.980 | 0.981 | 0.980 | 1.00 | 1.00 | 1.00 |
| S | 0.406 | 0.407 | 0.406 | 1.00 | 1.00 | 1.00 |
| Pm | 1.000 | 1.000 | 1.000 | 1.00 | 1.00 | 1.00 |
| Pe | 1.000 | 1.000 | 1.000 | 1.00 | 1.00 | 1.00 |
| Pt | 1.061 | 1.061 | 1.061 | 1.00 | 1.00 | 1.00 |
| Pq | 1.000 | 1.000 | 1.000 | 1.00 | 1.00 | 1.00 |
| Px | 1.000 | 1.002 | 1.001 | 1.00 | 1.00 | 1.00 |
| Pd | 1.000 | 1.000 | 1.000 | 1.00 | 1.00 | 1.00 |
| Er | 1.000 | 1.000 | 1.000 | 1.00 | 1.00 | 1.00 |
| Z | 0.383 | 0.383 | 0.383 | 1.00 | 1.00 | 1.00 |
| Sg | 0.086 | 0.087 | 0.087 | 1.01 | 1.01 | 1.00 |

Source: Computed.

As can be seen from the table, the percentage difference of the solution values provided by the solver program from the original values is less than the tolerable level of plus or minus 5 percent value for each of the endogenous variables, implying that the model is accurate. The initial (observed) values will then be replaced with the solution values as base year values of the endogenous variables. From this, the robustness test of the model follows.

### 6.2. Robustness of the Model

As is the case with most CGE model-based studies, key parameters of the RDCGE model are estimated based on educated guess rather than the econometrics method. Calibration method is used to estimate the remaining parameters. While calibration method allows one to estimate the model with only one period data, it is often criticized as one cannot objectively test the robustness of the parameter estimates and, thus, the simulation results. Therefore, the model needs to undergo a sensitivity test to assess whether the simulation result is significantly affected in case the true parameters differ from our estimates.

The sensitivity test is conducted by significantly changing both the value of export transformation elasticity and the import substitution elasticity, solving the model, and then comparing the original and solution values of the endogenous variables. Accordingly, the sensitivity of the model is tested using two scenarios.

As seen from the above table, the solution values for the first scenario and the solution values for the second scenario are identical (column F of the table above). When they are compared with the respective initial or original values, the result shows that the solution values are equal to their respective original values, as shown in columns D and E of the above table. Accordingly, the model is not sensitive to changes in the elasticity of export transformation and elasticity of import substitution such that the model is robust, dependable, and stable for simulation analysis.

### 6.3. Model Simulation

The model is then simulated under two import tariff reduction scenarios for the coming 30 years, from 2013/2014 to 2043/2044. In each case, the model simulation result regarding the major macroeconomic indicators in general and that of the fiscal indicators (including government tax revenue, government expenditure, the government primary balance, and other fiscal variables) are computed and recorded. Based on the model outputs, the possible impacts of the import tariff reduction on the major macroeconomic variables will be analyzed under each scenario. Moreover, the impact of import tariff reduction as compared to the no-tariff reduction measure is analyzed.

### 6.4. Impact on Price

The domestic import price (Pm) is a positive (an increasing) function of the World import price (Wm), the average import tariff rate (tm), and the exchange rate (Er). Given the world price of imports and the exchange rate remains constant, the domestic price of imports changes following the changes in the average import tariff rate. Accordingly, the average import price (Pm) remains unchanged under the base case scenario, as the import tariff rate remains unaffected. It, however, decreases by about 4.5 percent (to about 0.955 Birr) following the 95 percent reduction in the average import tariff. It decreases to about 0.955 Birr in the first year and remains unchanged over the remaining years under the 95 percent import tariff reduction scenario. This means that the import tariff reduction has a negative dynamic impact on the average import price.

### 6.5. Fiscal Sustainability

The Engel–Granger cointegration test [18] (under 95 percent reduction) proves that the primary balance ratio (pbt1) and the one-period lagged value of the government debt stock ratio (dt1) are nonintegrated. Therefore, the fiscal policy is not sustainable in the long run under the 95 percent import tariff reduction.

### 6.6. Average Consumption Tax

The results revealed that the average consumption tax rate should be raised by 23 percent so as to compensate for the tax revenue lost due to the introduction of the 95 percent import tariff reduction.

### 6.7. Impact on Macroeconomic Variables

The impact of the 95 percent import tariff reduction on all nominal variables is negative. Accordingly, the impact of a 95 percent import tariff reduction on aggregate saving (S), government saving (Sg), tax revenue (Tax), household income (Y), government fiscal deficit (Pbt), and government debt stock ratio is all negative. However, the impact on real variables is mixed. The impact on export (E), import (M), the quantity of composite commodity supply (Qs), and household consumption (Cn) is positive. However, the rate by which each real variable increases and the trend performance of each varies. However, the impact of the 95 percent import tariff reduction measure on aggregate investment and on the supply (Ds) and demand (Dd) for domestically produced output is negative.

## 7. Conclusions and Policy Implications

This study tried to link the dynamic CGE model developed with the fiscal reaction function to analyze how the macroeconomic effects on major economic variables from the tariff policy change transitively affect the sustainability of fiscal policy.

The analysis result indicates that the impact of import tariff reduction on price and nominal variables is negative. Although its effects on most real variables are positive, its negative impacts on prices exceed its positive impacts on the real variables such that the net effect on nominal variables is negative. In particular, the impact of the 95 percent tariff reduction on aggregate investment, a real variable, is negative. The impact of import tariff reduction is positive on the primary balance (fiscal deficit), meaning the deficit increases or widens under the tariff reduction. Its effect on government debt stock is also positive, implying that the debt accumulates more under the tariff reduction than in the base case scenario. Moreover, the effect of the import tariff reduction on the sustainability of fiscal policy is negative such that the sustainability is lost under the import tariff reduction scenario.

The analysis of the level to which the domestic average consumption (indirect) tax rate should be raised so as to maintain tax revenue and fiscal sustainability reveals that the domestic consumption tax rate should be raised by 23 percent, given the 95 percent import tariff reduction measure. The possible effects of the joint reform system on major variables negatively affected under the import tariff reduction measure are also analyzed. The analysis result shows that the joint reform improves mainly nominal variables, including government tax revenue, government saving, aggregate saving, narrow fiscal deficit, and decreased government debt stock accumulation. It also positively affected aggregate investment, negatively affected by the import tariff reduction policy alone. In sum, mainly in countries where governments play a significant role in the economy, trade policies with negative consequences on government tax revenue, such as the import tariff reduction, should be complemented with fiscal policies that could maintain tax revenue and the sustainability of fiscal policy.

### 7.1. Policy Implications

The policy recommendations follow from the research findings. Ethiopia should improve its domestic economic performance by improving the competitiveness of its exports. It should also increase its competitiveness in import production. Tax management needs to become efficient and productive to reduce tax evasion. More importantly, a phase-wise tariff reduction is advisable rather than a single significant reduction. Deficit financing, concessional borrowing prioritized expenditures are needed. Further liberalization and promotion of foreign direct investment flows are required. On the foreign grants front, systematic and efficient diplomatic efforts must be initiated.

### 7.2. Suggestions for Further Research

The analysis of the impacts of import tariff reduction on macroeconomic variables and the long-run sustainability of fiscal policy is performed under several restrictive assumptions. From among the assumptions made, a constant annual 3 percent decrease in government consumption expenditure GDP ratio, an annual 1 percent increase in output GDP ratio, one-time implementation of the 95 percent import tariff reduction, etc. are some.

Future research can analyze the impacts of the import tariff reduction on different macroeconomic variables and the sustainability of fiscal policy by relaxing some of the assumptions made by this research. Therefore, some of the issues that could be addressed by future research are indicated below.

Analyzing the impacts of a phase-wise (gradual) import tariff reduction (say reducing import tariff by 95 percent over the coming 15 years in three phases) on different macroeconomic variables and the fiscal sustainability of Ethiopia could be one area of the future.

**Author Contributions:** G.R. and B.A.G. co-designed the research, collected the data, drafted the manuscript, and conducted the statistical analyses and drafted the manuscript. C.P. formulated the research questions and focus, co-designed the research, and drafted the manuscript with the first two authors. R.J. and S.K. providing important ideas and substantial feedback to the study. All authors have read and agreed to the published version of the manuscript.

**Funding:** This research received no external funding, and the APC was funded/contributed by the authors.

**Institutional Review Board Statement:** Not applicable.

**Informed Consent Statement:** Not applicable.

**Data Availability Statement:** Available at the ministries' websites in Ethiopia.

**Acknowledgments:** The authors would like to thank the editor and all the anonymous referees for their excellent comments and suggestions, which were most helpful in revising the paper.

**Conflicts of Interest:** The authors declare no conflict of interest.

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
