# Peer review of "Import Tariff Reduction and Fiscal Sustainability: A Macro-Econometric Modelling for Ethiopia"

_sustainability, doi:10.3390/su15043074_

Round 1

Reviewer 1 Report

Dear Editor,

Thank you for giving me the opportunity to review the paper entitled "Import Tariff Reduction and Fiscal Sustainability: A Macro Econometric Modelling for Ethiopia", with Manuscript ID: sustainability-2092603.

I have completed reading the paper. This is an interesting paper. The aims of this paper are as follows: (1) Estimation of the dynamic impact of import tariff reduction on significant macroeconomic variables, and (2) Estimation of increasing in the average consumption tax required to compensate for the possible tax revenue loss and fiscal sustainability.

However, based on the following comments, I believe this paper must be revised before publication in sustainability.

Thus, I recommend revising this paper.

Yours sincerely,

Comments:

1. The authors have not explained how to obtain Equations (1) to (23). On other words, if the authors have proposed these equations, they must clarify how to obtain them. But, if these equations have been already proposed by other researchers, the authors should cite thier related references in the text.

2. The innovation of this paper is not clear. The authors should illustrate the innovation so that the readers can find it in the paper easily.

3. The authors should explain the managerial and economic implications of this paper as a new section.

4. The authors should compare their obtained results with the results obtained from the previous related studies.

5. The English language of this paper should be revised carefully.

Author Response

Respected Sir/Madam,

Thanks for your very constructive comments and suggestions. We have thoroughly revised the document per your suggestions and to the best of our capabilities. Please find changes in the attached document. 

Reviewer 2 Report

Major

1.      Authors should emphasize the novelty of the current research. Not only because no such studies are available on Ethiopia, but also the modeling approach, the unique model features, the numerical method, etc.

2.      In the literature study, authors should add some more relevant and recent references reviewing the Recursive Dynamic Computable General Equilibrium Model (RDCGEM): the advantages and limitations as well as its application in economic modeling. The position of the current manuscript relative to other studies, especially in the similar topic, should also be highlighted.

3.      As mentioned in the introductory part, the objective of the study is to measure the effect of import tariff reduction on import, export, current account balance, government tax, fiscal sustainability, and domestic sales tax. However, the Discussion section is very limited by focusing the discussion only on the impact of import price. Authors must expand the Discussion section such that it includes the responses to the research questions.

Minor

1.      The first paragraph of Introduction must be dropped since it’s redundant with the second paragraph.

2.      The way of writing parameters, variable, and mathematical equations should be improved following the common styles (italic, subscript, small/capital, etc.).

3.      Is there any reason on why the model was calibrated using 2013/14 base year macroeconomic data?

4.      The export transformation parameters in equation (1) and in description have different symbols. So does the Armington parameter in equation (3).

5.      Equation (2) must be corrected.

6.      Pbt and I_g in equation (11) should be introduced.

Author Response

Respected Sir/Madam,

Thanks for your very constructive comments and suggestions. We have thoroughly revised the document per your suggestions and to the best of our capabilities. Please find attached documents:

Reviewer 3 Report

  1. Abstract- It needs to highlight the aim of this study. The first two sentences only mention the study's methodology, which does not give a clear indication of the purpose of this research.
  2. This is similar to the introduction section, where the authors seem very excited to highlight the methodology used in this research.
  3. This manuscript would be better if it started to discuss the background of this research study, which is import tariff reduction.
  4. Once again, in the literature review, the authors seem too excited with the methodologies rather than discussing the significance and role of import tariff reduction, especially in Ethiopia. 
  5. The literature review also needs to explain the other important subject matter of this research, fiscal sustainability. This is important, especially to fit the manuscript in the journal.
  6. The importance of discussing the import tariff, which is the main research background, is that the authors need to develop the narrative story to establish the case study of this research.
  7. The explanations of the data need to be clarified. Please describe the type of data together with the source.
  8. "Secondary data is collected from the Ministry of Finance and Economics Coop- 197 Corporation (MoFEC), the Planning Commission, the Central Statistical Agency (CSA), 198 and the National Bank of the Ethiopian Federal Democratic Republic (EFDR). Other 199 data sources are the World Bank database, the WTO database, the IMF database, 200 previous research documents, and other previous surveys"- This paragraph needs to be further discussed.
  9. Section 3 is misleading. It is not data analysis but Research Methodology.
  10. In explaining data methodology, authors also need to discuss it within the theoretical approach. That means authors need to relate similar techniques in other studies that aim to generate similar outputs.
  11. The discussion of the findings is way too simple. Authors can also discuss the results with some relevant Ethiopian economic and international policies related to the import tariff.
  12. Besides, authors should also discuss the findings within a similar scope in economics studies, especially on import tariffs. It is not just based on local studies but also on other countries, especially developed ones.
  13. More importantly, the discussion also did not relate to fiscal sustainability. 
  14. Overall this paper is an exciting topic and will significantly contribute to the local economy. Nevertheless, in term of discussion still need significant improvement.
  15. The research methodology is well explained.
  16. The findings also need significant improvement, especially to relate with the important subject matters of this research: import tariff and its relationship to fiscal sustainability

Author Response

Respected Sir/Madam,

Thanks for your very constructive comments and suggestions. We have thoroughly revised the document per your suggestions and to the best of our capabilities. Please find the attached of revision of the paper

Thank you

Paramaiah

Reviewer 4 Report

Import Tariff Reduction and Fiscal Sustainability: A Macro Econometric Modelling for Ethiopia

This paper has the declared objectives of assessing the impact of a tariff reduction in Ethiopia on macroeconomic variables, including fiscal sustainability. It also envisages to calculate the tax increase on consumption needed to offset the fiscal impact of the tariff reduction. For that purpose, the authors use a Dynamic Computable General Equilibrium Model.

The data used in the model is however quite unclear. The authors do not mention the period, only mentioning that 2013-2014 was used as a base year. They say afterwards that the model is simulated under two import tariff reduction scenarios for the coming 30 years from 2013/14 to 2043/44. From that statement it may be inferred that the research was performed some time ago. It is also not clear if the import tariff reduction actually took place or it is simulated.

The result also point that a large tariff reduction leads to fiscal unsustainability. It would be relevant to show the impact of such a cut as a percentage of GDP. It seems like a hard conclusion and more proof is needed in order to support that.

As a general comment, there is a need to improve the clarity of the paper. What is the actual situation in Ethiopia related on tariffs, what are the hypothesis related to tariff reductions etc. Also, the research should use recent data for the estimations.  

With these elements clarified, the research could be interesting and relevant for policymakers.

Author Response

Respected Sir/Madam,

Thanks for your very constructive comments and suggestions. We have thoroughly revised the document per your suggestions and to the best of our capabilities. 

Thank you 

Paramaiah

Round 2

Reviewer 1 Report

Dear Editor, Thank you for giving me the opportunity to review the revised paper entitled “Import Tariff Reduction and Fiscal Sustainability: A Macro Econometric Modelling for Ethiopia”, (with Manuscript ID: sustainability-2092603).
I have completed reading the revised paper. In my opinion, the authors have addressed the points raised in the previous review with patience.
I believe that the revised paper is now acceptable for publication in sustainability.
Yours sincerely,

Author Response

Dear Sir/Madam, 

Please find the attached after incorporating your suggestions and comments. 

With Regards

Paramaiah

Reviewer 3 Report

Overall there has been a significant improvement from the previous version. Nevertheless, I would like to suggest some edits for the following aspects: 

  1. The Abstract has been improved
  2. The introduction section needs to give some overview of the case study, especially an overview of the macroeconomic situation in Ethiopia. 
  3. The introduction section also needs to explain the relationship between the macroeconomic situation and import tariff reduction.

Author Response

(The authors gave the same response as above.)

Reviewer 4 Report

The authors efforts to improve the paper are appreciated.

However, neither in the letter to the reviewer or in the text of the paper I couldn’t find an answer to the majority of my concerns.

First, what is the actual situation in Ethiopia related on tariffs, what are the hypothesis related to tariff reductions etc. The research apparently used quite old data (2013-2014 base year).  It is not clear why and if some kind of tariff reduction took place until the present or we are discussing simulations.

Second, the statement that this tariff reduction leads to fiscal unsustainability is quite hard and has not been proven. First, the impact of the tax reduction should be measured (preferable as a percentage of GDP) and then investigated if this decrease in budgetary revenues leads to fiscal unsustainability. The authors did mention some articles related to fiscal sustainability, but the methodology used for that purpose is limited. Also, the authors should isolate the impact of the tariff reduction only and not investigate directly cointegration between primary balance and public debt – it could be the case that fiscal unsustainability is present irrespective of the tariff reduction. This objective of the paper is not therefore met in my view and should be tackled with a significant improved methodology.

Third, I reiterate the need to improve the clarity of the paper. Clear stating of the objectives, of the data used, of the methodology, of the pertinence of the results are mandatory.  

In my view, the paper needs a significant redrafting.

Author Response

(The authors gave the same response as above.)

Round 3

Reviewer 4 Report

I thank the authors for their answers.

However, in my view, the changes operated to the manuscript are not very comprehensive.

In this form, I don’t believe that the objectives of the paper are met. The first objective was to examine the impact of import tariff reduction on Ethiopia's tax revenue. I couldn’t find that in the results section. I suggested previously that this impact should be quantified as a percentage of GDP. Another objective of the paper was to assess the impact of import tariff reduction on Ethiopia's fiscal sustainability. I consider that this objective is also not met. The impact of the tax reduction should be measured and then investigated if this decrease in budgetary revenues leads to fiscal unsustainability. The authors should isolate the impact of the tariff reduction only and not investigate directly cointegration between primary balance and public debt – it could be the case that fiscal unsustainability is present irrespective of the tariff reduction.

In addition, there are some other elements related to the clarity of the paper which should be improved.